

# Production of oxygenated volatile organic compounds from the ozonolysis of coastal seawater

Delaney B. Kilgour[1], Gordon A. Novak[1, ǂ], Megan S. Claflin[2], Brian M. Lerner[2], Timothy H. Bertram[1]

[1]Department of Chemistry, University of Wisconsin-Madison, Madison, 53706, USA
[2]Aerodyne Research Inc., Billerica, 01821, USA
[ǂ]Now at: NOAA Chemical Sciences Laboratory, Cooperative Institute for Research in Environmental Sciences, University of Colorado, Boulder, 80305, USA

*Correspondence to*: Timothy H. Bertram (timothy.bertram@wisc.edu)

**Abstract.** Dry deposition of ozone ($O_3$) to the ocean surface and the ozonolysis of organics in the sea surface microlayer
(SSML) is a potential source of volatile organic compounds (VOC) to the marine atmosphere. We use a gas chromatography
system coupled to a Vocus proton transfer reaction time-of-flight mass spectrometer to determine the chemical composition
and product yield of select VOC formed from ozonolysis of coastal seawater collected from Scripps Pier in La Jolla,
California. Laboratory-derived results are interpreted in the context of direct VOC vertical flux measurements made at
Scripps Pier. The dominant products of laboratory ozonolysis experiments and the largest non-sulfur emission fluxes
measured in the field correspond to Vocus $C_xH_y^+$ and $C_xH_yO_z^+$ ions. GC analysis suggests that $C_5$-$C_{11}$ oxygenated VOC,
primarily aldehydes, are the largest contributors to these ion signals. In the laboratory, using a flow reactor experiment, we
determine a VOC yield of 0.43-0.62. In the field at Scripps Pier, we determine a maximum VOC yield of 0.04-0.06. Scaling
the field and lab VOC yields for an average $O_3$ deposition flux and an average VOC structure results in an emission source
of 12.6 to 136 Tg C $yr^{-1}$, competitive with the DMS source of 21.1 Tg C $yr^{-1}$. This study reveals that $O_3$ reactivity to
dissolved organic carbon can be a significant carbon source to the marine atmosphere and warrants further investigation into
the speciated VOC composition from different seawater samples, and the reactivities and secondary organic aerosol yields of
these molecules in marine-relevant, low $NO_x$ conditions.

## 1 Introduction

The ocean surface acts as a source of volatile organic compounds (VOC) to the atmosphere (Carpenter et al., 2012), with
subsequent impacts on oxidant concentrations and the production of secondary organic aerosol (SOA) (Donahue and Prinn,
1990; Meskhidze et al., 2011; Rinaldi et al., 2010). Three marine VOC production pathways have been proposed: (1)
biogenic production, controlled by marine biological processes (Carpenter et al., 2012), and abiotic production via (2)
photochemical (Brüggemann et al., 2018; Carpenter and Nightingale, 2015; Ciuraru et al., 2015a; Novak and Bertram, 2020)
or (3) heterogeneous oxidation reactions at the surface ocean (Carpenter and Nightingale, 2015; Novak and Bertram, 2020;
Zhou et al., 2014).





To date, marine VOC research has largely focused on biogenic VOC (BVOC), with primary attention to dimethyl sulfide (DMS) and to a lesser extent, isoprene and monoterpenes (Lana et al., 2011; Shaw et al., 2010). Marine DMS emissions are estimated to be 13.2-25.8 Tg C yr$^{-1}$ (Lana et al., 2011), while marine isoprene and monoterpene emissions are estimated to be

0.1-12 Tg C yr$^{-1}$ and 0.01-29.5 Tg C yr$^{-1}$ (Luo and Yu, 2010), respectively, but generally considered to be on the lower end of these estimates. This collection of VOC has well-established biological production pathways and impacts on secondary marine aerosol production and cloud properties (Bates et al., 1992; Charlson et al., 1987; Kiene and Linn, 2000; Shaw et al., 2010). While isoprene and monoterpene emissions of carbon are much smaller than DMS emissions, they have high SOA yields and much faster bimolecular reaction rate constants with ozone (O$_3$) and hydroxyl radicals (OH) than DMS, leading to

outsized atmospheric impacts despite their smaller emission rates (Griffin et al., 1999; Kroll et al., 2006; NASA Jet Propulsion Laboratory, 2020).

Within the last ten years, several laboratory experiments and field campaigns have provided evidence that marine VOC can be formed through abiotic mechanisms at the air-sea interface sea surface microlayer (SSML) (Chiu et al., 2017; Ciuraru et

al., 2015a, b; Coburn et al., 2014; Fu et al., 2015; Mungall et al., 2017; Penezić et al., 2023; Schneider et al., 2019; Zhou et al., 2014). Measurements of the SSML have shown it is enriched in unsaturated organics that can undergo photochemical or heterogeneous oxidation reactions with oxidants like O$_3$ deposited to the ocean surface, and its presence is widespread across the global ocean at wind speeds up to 10 m s$^{-1}$ (Wurl et al., 2011). Laboratory experiments using model and authentic SSML monolayers have shown photochemical production of a collection of saturated and unsaturated reactive compounds (Chiu et

al., 2017; Ciuraru et al., 2015a, b; Fu et al., 2015). Modelling studies suggest interfacial photochemistry could source up to 91.9 Tg C yr$^{-1}$ (Brüggemann et al., 2018).

The deposition velocity of O$_3$ to the ocean surface is thought to be controlled by the reaction rates of O$_3$ with dissolved iodide (I$^-$) and dissolved organic carbon (DOC). The reaction of O$_3$ with I$^-$ leads to the production of iodine (I$_2$) and

hypoiodous acid (HOI) (Carpenter et al., 2021), while the reaction of O$_3$ with DOC can lead to the production of VOC and oxygenated VOC (OVOC) (Chang et al., 2004; Schneider et al., 2019; Wang et al., 2023; Zhou et al., 2014). However, the concentration and composition of DOC in the SSML and the reaction kinetics for O$_3$ + DOC species are not well-constrained (Clifford et al., 2008). It is estimated that the reaction rate constant for O$_3$ + I$^-$ is orders of magnitude faster than for the O$_3$ + DOC reaction, where $k_{O_3 + I^-} = 2.4 \times 10^9 \, M^{-1}s^{-1}$ at 293 K (Magi et al., 1997), leading to an O$_3$ reactivity on the order of

100-300 s$^{-1}$ for representative oceanic I$^-$ concentrations (40-120 nM) (Chance et al., 2019), and $k_{O_3 + DOC} = 2.6 \times 10^7 \, M^{-1}s^{-1}$ for a marine DOC sample (Shaw and Carpenter, 2013). Due to the paucity of authentic marine $k_{DOC}$ measurements, estimates for O$_3$ reactivity to DOC are relatively unknown, with the only measurement on authentic marine DOC reporting a value of 1820 ± 560 s$^{-1}$ for 7 × 10$^{-5}$ M DOC (Shaw and Carpenter, 2013). In coastal regions, it is thought that O$_3$ reactivity to DOC is large enough to be competitive with I$^-$, whereas I$^-$ dominates O$_3$ reactivity in the open ocean

(Shaw and Carpenter, 2013; Ganzeveld et al., 2009). Despite uncertainties in the reactivity of $O_3$ with DOC, laboratory studies have shown it is high enough to support the production of a variety of VOC. Zhou et al. (2014) showed that model SSML containing linoleic acid and authentic SSML samples, when exposed to > 350 ppb $O_3$, resulted in prompt emission of a variety of aldehydes at high yield via reaction at the substrate's carbon-carbon double bond and subsequent decomposition of the primary ozonide (Zhou et al., 2014). Similarly, Schneider et al. (2019) measured $C_1$, $C_5$, and $C_7$-$C_{10}$ carbonyl products

from ozonolysis (at 8.5 ppm) of an authentic SSML created from a phytoplankton culture (Schneider et al., 2019). Most recently, Wang et al. (2023) performed ozonolysis experiments with roughly 100 ppb $O_3$ on 10 SSML samples from the South China Sea and reported production rates of acetaldehyde, acetone and/or propanal, and $C_6$-$C_9$ saturated aldehydes.

Using an average $O_3$ deposition flux ($1.5 \times 10^{10}$ molecules $cm^{-2}$ $s^{-1}$ corresponding to an $O_3$ concentration of 30 ppb and

deposition velocity of 0.02 cm $s^{-1}$), Novak and Bertram (2020) estimated that the carbon mass flux of VOC from ozonolysis of the seawater surface to be 17.5-87.3 Tg C $yr^{-1}$ (for $\varphi_{VOC}$ ranging 0.1-0.5) (Novak and Bertram, 2020), competitive with the carbon mass flux from BVOC and a proposed photochemical source (Brüggemann et al., 2018; Lana et al., 2011). Importantly, the set of molecules produced from the ozonolysis of seawater can be larger, more oxygenated, and unsaturated compared to common marine BVOC, like DMS and isoprene (Schneider et al., 2019; Wang et al., 2023; Zhou et al., 2014).

These properties can enable these molecules to be efficient precursors of SOA and cloud condensation nuclei (CCN) (Lim et al., 2019; Zhao et al., 2015). However, our ability to determine the extent to which this abiotic pathway is active over the ocean and its corresponding atmospheric impacts is limited by uncertainty in how to bridge the gap between fundamental laboratory experiments of ozonolysis with model SSML and the significantly more complex and spatially variable seawater surface in ambient environments.


Here we present field measurements of direct eddy covariance VOC vertical fluxes at Ellen Browning Scripps Memorial Pier (herein Scripps Pier) in La Jolla, CA collected with a high resolution Vocus proton transfer reaction time-of-flight mass spectrometer. We assess the contribution of abiotic heterogeneous oxidation from ozonolysis to the measured carbon mass emission flux at Scripps Pier through controlled laboratory seawater ozonolysis experiments sampled with a coupled gas

chromatography system.

Our results indicate that the yield of VOC from seawater ozonolysis at the ocean surface ($\varphi_{VOC}$) is at maximum 0.06, while the yield determined in flow reactor experiments in the laboratory are as large as 0.62. Even at the low yield limit ($\varphi_{VOC}$ = 0.06), the ozonolysis of surface seawater is expected to be a significant source of reactive carbon to the marine atmosphere.



## 2 Methods

### 2.1 Seawater collection, storage, and measurements

Seawater used in the laboratory experiments was pumped from below Scripps Pier in La Jolla, CA (32-52'00" N, 117-15'21" W) on 11 November 2020. Collected seawater was filtered through 50-micron Nitex nylon mesh (Flystuff, Cat#57-106) and stored in amber 1 L Nalgene HDPE bottles. The water was shipped frozen and stored in a -20 ºC freezer before and after shipping on 16 November 2020. Individual aliquots of seawater were defrosted to room temperature immediately prior to use in experiments, with all ozonolysis experiments completed within a year of sample collection. Seawater I$^-$ and DOC concentrations were measured by ion chromatography inductively coupled plasma mass spectrometry (IC-ICP-MS; Thermo Scientific ICS-2100 IC and Thermo Scientific iCAP RQ ICP-MS) and a total organic carbon analyzer (Sievers M5310C), respectively.

### 2.2 Flow tube experimental design

All laboratory experiments were performed in a flow tube assembled from a quartz glass tube (Technical Glass Products) and 316 stainless steel end plates. The flow tube had an inner diameter of 135 mm and a length of 122 cm, providing a total internal volume of 17.4 L. Each stainless-steel end cap was made with Swagelok fittings for headspace gas flow (Fig. S1). Prior to filling with seawater, $O_3$ was passed through the flow tube for 60 minutes to oxidize residual contaminants adhered to the walls and provide a clean headspace for the seawater experiments. $O_3$ was generated by passing 100 sccm ultra-high purity (UHP) $O_2$ (OXUHP300, Airgas) and 3900 sccm UHP $N_2$ (NIUHP300, Airgas) over a UV lamp (254 nm Pen-Ray Lamp, Jelight, Inc.), producing an $O_3$ concentration of 90 ppb measured with a commercial ozone monitor (Model 49i Ozone Analyzer, Thermo Fisher Scientific).

Once the flow tube was cleaned, it was filled with 1 L of water for each experiment. After filling with water, 4000 sccm UHP air (3200 sccm UHP $N_2$ and 800 sccm UHP $O_2$) and 1.4 sccm UHP $CO_2$ (CD UP300, Airgas), resulting in 350 ppm $CO_2$, flowed through the headspace for approximately 45 minutes, allowing for BVOC to degas and the SSML to establish. $CO_2$ was added to improve $O_3$ detection by the chemical ionization mass spectrometer (Novak et al., 2020) described in Sect. 2.3. After 45 minutes, flow switched to 4000 sccm 90 ppb $O_3$ in $N_2$ to probe VOC produced from ozonolysis of the surface water, shown in the schematic in Fig. S2. Both setups led to an average residence time of air in the flow tube of 4.3 minutes. These experiments were also performed using Milli-Q water, which served as a blank for VOC emissions and $O_3$ deposition. Details of the experimental configuration are presented in Table S1 of the supplementary information.

### 2.3 Laboratory VOC and O₃ measurements

A high-resolution Vocus proton transfer reaction time-of-flight mass spectrometer (PTR-TOF-MS) (herein referred to as RT-Vocus to denote its real-time (RT) operation) (TOFWERK, Aerodyne Research, Inc.) made continuous measurements of



VOC (19-500 *m/Q*) at 1 Hz time resolution and with a mass resolution of *~4000 m/Δm* (Krechmer et al., 2018). The focusing ion-molecule reactor had a temperature of 100 ºC, pressure of 1.5 mbar, and axial electric field gradient of 36.5 V cm$^{-1}$, leading to a reduced electric field strength (E/N) of 125 Td.

A gas chromatography (GC) system equipped with *in situ* thermal desorption preconcentration (Aerodyne Research, Inc.) was used in tandem with the Vocus, referred to as GC-Vocus, to speciate isomers and determine parent molecules of observed fragment ions (Claflin et al., 2021; Vermeuel et al., 2023). The GC-Vocus preconcentrated analytes by collecting 1 L of air over a 10-minute sampling period through a heated sodium sulfite (Na$_2$SO$_3$) oxidant trap at 35 ºC and into a thermal desorption pre-concentrator (TDPC). The oxidant trap served to remove O$_3$ from the sample air to prevent degradation of the

adsorbent traps and column. In the TDPC, sample is first collected onto a multi-bed adsorbent trap (Tenax TA/Graphitized Carbon/Carboxen 1000, Markes International) and then is transferred to a multi-bed cold trap (Tenax TA/Carbpoack X/Carboxen 1003, Markes International), both held at a temperature of 20 ºC. The sample flow is next injected onto the GC column (MXT-624, Restek) which follows a programmed temperature ramp from 35-225 ºC. This combination of adsorbents and column allows for the detection of a wide range of VOC and OVOC, with the system optimized for the analysis of C$_5$-

C$_{12}$ VOC and C$_2$-C$_{10}$ OVOC. The GC was operated on a 30-minute total cycle which included a 10-minute sample collection period and a 20-minute chromatographic separation, recorded by the GC-Vocus at 5 Hz.

Both the GC-Vocus and RT-Vocus subsampled 100 sccm from approximately 3 m of 0.25″ O.D. PFA tubing that pushed 4000 sccm from the flow tube. Since the long times required for preconcentration and elution from the column are

challenging for fast-changing experiments such as these, individual experiments consisting of the entire sequence in Table S1 were sampled by either the RT-Vocus or by the GC-Vocus. Because the GC-Vocus and RT-Vocus did not sample simultaneously during these experiments, we use the GC measurements to qualitatively identify molecules rather than quantify molecules. The collection of RT-Vocus ions that responded to ozonolysis and their temporal signal response to ozonolysis was consistent with total VOC peaking within 6.5 and 7.5 minutes in all RT-Vocus experiments, lending

confidence that the experiments were reproducible and thus reproducibility was also assumed for comparison between RT-Vocus and GC-Vocus results.

A chemical ionization time-of-flight mass spectrometer (CIMS) (Aerodyne Research, Inc., TOFWERK) was operated with oxygen anion reagent ion chemistry (Ox-CIMS) to measure O$_3$ at 1 Hz (Bertram et al., 2011; Novak et al., 2020). The high

precision and time resolution of the Ox-CIMS was required to measure quick and small fluctuations in O$_3$ during the ozonolysis experiments. A brief explanation of the Ox-CIMS is reported below, with more details on the instrument and ionization scheme available in Novak et al. (2020). Oxygen anions were generated by flowing 2200 sccm UHP N$_2$ and 400 sccm UHP O$_2$ through a polonium-210 α-particle source (NRD, P-2021 Ionizer). Oxygen anions reacted with sample air in an ion-molecule reaction (IMR) chamber held at 95 mbar. The product ions then passed through three stages of differential



pressure before reaching the ToF mass analyzer. In the experimental conditions used for this study (85% RH, 350 ppm $CO_2$), $O_3$ was primarily detected at the $CO_3^-$ product ($m/Q$ $60$) (Novak et al., 2020). As a result, $O_3$ was measured as $CO_3^-$ normalized to the sum of the reagent ion ($O_2^-$) and first reagent ion water cluster ($O_2^-(H_2O)$). Since the detection of $O_3$ as the $CO_3^-$ product is dependent on the $CO_2$ concentration in the experimental flow (Novak et al., 2020), a Los Gatos Research Carbonyl Sulfide Analyzer was placed in-line to continuously measure $CO_2$ (Berkelhammer et al., 2016).


Peak fitting and integration of GC-Vocus, RT-Vocus, and Ox-CIMS data were completed in Tofware v3.2.3 (Aerodyne Research, Inc., TOFWERK). Chromatogram peak areas were determined using TERN v2.2.18 (Aerodyne Research, Inc.). A collection of non-methane VOC were calibrated on the RT- and GC-Vocus using a custom 14-component VOC calibration cylinder (Apel-Riemer Environmental, Inc.). Aldehyde molecules including pentanal, hexanal, heptanal, octanal, and

nonanal (all Millipore Sigma, >95% purity) were calibrated on the RT-Vocus by direct injection of aldehyde molecules diluted in methanol into UHP air carrier gas flow. All calibration factors were assumed to be insensitive to specific humidity (Krechmer et al., 2018). Because multiple molecules contributed to individual $C_xH_y^+$ ions, the average calibration factor (1.29 cps ppt$^{-1}$) of molecules to their largest product ions was applied to all RT-Vocus ions in Table S3. $O_3$ measured by the Ox-CIMS was quantified through humidity-dependent calibrations using a calibrated ozone source (Model 306 Ozone Cal

Source, 2B Technologies) and an ozone monitor (Model 49i Ozone Analyzer, Thermo Fisher Scientific).

## 2.4 Scripps Pier VOC and $O_3$ flux measurements

Continuous measurements of $O_3$ (via the Ox-CIMS) (Novak et al., 2020) and of VOC (via the RT-Vocus) (Novak et al., 2022) concentrations and eddy covariance vertical fluxes were made from the coast at Scripps Pier during Summer 2018 and September 2019, respectively. In both deployments, the instruments were contained in a temperature-controlled trailer at the

end of the 330 m long pier extending 100 m beyond the wave-breaking zone, and their sampling inlets were mounted on a 6.1 m long boom that extended beyond the pier. Details of the Ox-CIMS and RT-Vocus operations and inlet configurations are available in Novak et al. (2020) and Novak et al. (2022), respectively. Resolvable fluxes above the flux limits of detection were made for 43 ions corresponding to VOC in the RT-Vocus mass spectra, where the flux limit of detection was equivalent to an 80% confidence level. Of these 43 ions, 36 had a campaign average positive flux, indicative of emission.

The mean flux and flux limit of detection for the largest ion, $C_5H_9^+$, were 0.12 ppt m s$^{-1}$ and 0.08 ppt m s$^{-1}$, respectively.

## 3 Results and Discussion

## 3.1 Flux measurements of VOC emissions at Scripps Pier field site

Figure 1 presents hourly binned campaign diel averages of wind speed and fluxes for all 36 ions recorded on the RT-Vocus with a campaign average positive flux. For this analysis, we only utilize data collected with onshore winds (200-360º). The




number of flux measurements made during the night and early morning were limited and more variable due to wind primarily from the land during these times (Novak et al., 2022). Wind speeds showed a clear diel profile, peaking at 4 m s$^{-1}$ at 12 PDT local time, with the total carbon mass emission flux closely following wind speed throughout the day and night. The emission flux of molecules measured as ions on the RT-Vocus is translated to a carbon mass emission flux using the calibration factor for the expected molecule at each ion. Each ion is treated as a unique molecule, and thus the total carbon

mass emission flux should be interpreted as an upper limit. Ions without an expected molecule or calibration factor are quantified using the calibration factor of aldehydes in lab scaled to field sensitivities using the DMS calibration factor as a transfer standard. Details on the assignment of field calibration factors and a discussion of the introduced uncertainties are in Tables S2 and S3 and Supplemental S1, respectively. We note that several ions, including the $C_5H_9^+$ ion, often had a positive zero and have chosen not to background correct the data. The calculation of their emission fluxes, reported below, is

unaffected.

We show that approximately half of the carbon mass emission flux at the site is carried by organic material, primarily measured at ions with the formula $C_xH_y^+$, including $C_5H_9^+$ and $C_{10}H_{17}^+$, and $C_xH_yO_z^+$. $C_5H_9^+$ is the molecular ion of isoprene and a potential fragment for several larger molecules (Ruzsanyi et al., 2013; Pagonis et al., 2019), and $C_{10}H_{17}^+$ is the molecular ion of monoterpene isomers (Pagonis et al., 2019). Other ions with the formula $C_xH_y^+$ could represent

hydrocarbons or dehydrated products of alcohols or carbonyls, which could have biogenic or abiotic sources (Kim et al., 2010). $C_xH_yO_z^+$ ions similarly represent oxygenated organic products of either biogenic or abiotic origin. The remaining half of the carbon flux is composed of sulfur-containing molecules measured at $C_2H_7S^+$, corresponding to DMS, and $CH_5S^+$, corresponding to methanethiol (Novak et al., 2022). The small observed emission flux of Si and N-containing ions is attributed to the coastal nature and urban influence of this flux site (Coggon et al., 2018; Franklin et al., 2021).







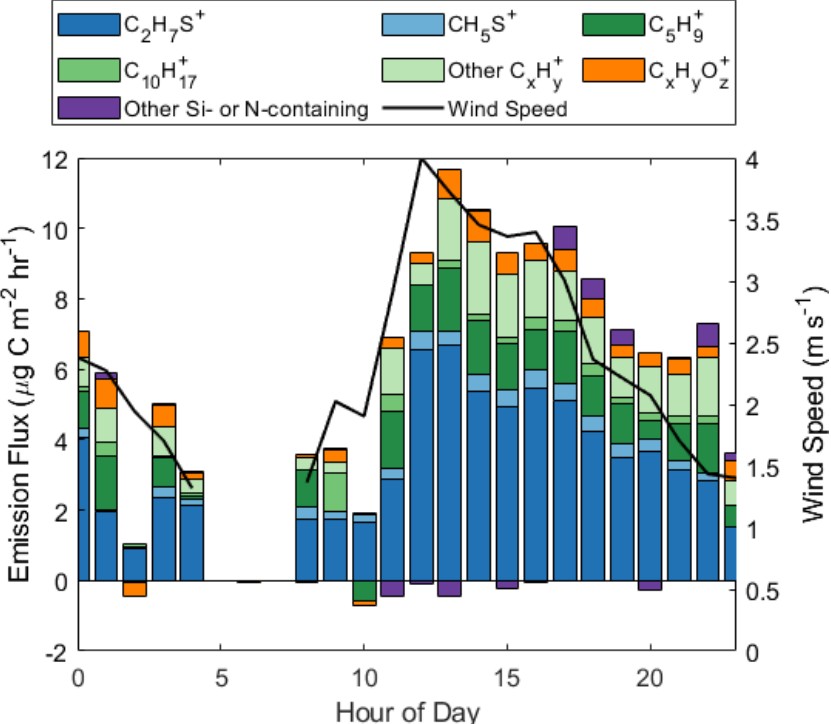

**Figure 1: Hourly binned wind speed and emission flux of all ions on the RT-Vocus with a campaign average positive flux, reported in units of µg C m$^{-2}$ hr$^{-1}$. Ions are grouped by chemical formula, where C$_x$H$_y^+$ includes all ions with the formula C$_x$H$_y^+$, excluding C$_5$H$_9^+$ and C$_{10}$H$_{17}^+$.**

C$_5$H$_9^+$ is the ion carrying the largest fraction of organic material. Furthermore, this ion accounted for up to 30% of the total carbon mass emission flux (at 8 PDT) and 12% on average for the campaign. The hourly binned isoprene-equivalent flux of C$_5$H$_9^+$ ranged between -0.56 and 1.8 µg C m$^{-2}$ hr$^{-1}$, with a campaign average of 1.1 µg C m$^{-2}$ hr$^{-1}$ or 0.12 ppt m s$^{-1}$. Other C$_x$H$_y^+$ ions (excluding C$_5$H$_9^+$ and C$_{10}$H$_{17}^+$) comprised maximum 23% of the total carbon mass emission flux (at 22 PDT) and 14% on average for the campaign.

Motivated by the large contribution of C$_5$H$_9^+$ ions to the total carbon emission flux at this site, we further investigated the likely molecular composition of this ion and its sources. Utilizing the Johnson and Nightingale parameterizations for transfer velocity with the measured wind speeds and emission fluxes at Scripps Pier, we can calculate what dissolved VOC concentration is needed to sustain observed fluxes (Johnson, 2010; Nightingale et al., 2000). For a wind speed of 3 m s$^{-1}$ and an emission flux of 0.12 ppt m s$^{-1}$, we calculate a dissolved isoprene concentration of 0.74 nM is needed to sustain the observed C$_5$H$_9^+$ emission flux if this ion is solely isoprene. This dissolved concentration is at least an order of magnitude larger than typical oceanic dissolved isoprene concentrations of 1-100 pM (Shaw et al., 2010; Hackenberg et al., 2017; Li et





al., 2022). Furthermore, in GC-Vocus experiments measuring the headspace of degassing seawater collected from this site, no isoprene was observed despite observations of other degassing BVOC, like DMS (Fig. S3).

It has also been suggested that large isoprene fluxes could be driven by photochemistry in the SSML (Brüggemann et al., 2018; Ciuraru et al., 2015b), which would not require a dissolved isoprene concentration. Our observations do not support a photochemical source of $C_5H_9^+$, as no correlation between $C_5H_9^+$ flux and solar irradiance was observed (Fig. 2). This follows observations in the North Atlantic (Kim et al., 2017) where the same finding was shown but for a more limited irradiance range (0-500 W m$^{-2}$). However, the averaged $O_3$ deposition flux at this site is large enough that even a small

product yield from ozonolysis could support these observations. While no clear dependence of the $C_5H_9^+$ flux on $O_3$ concentrations is measured (Fig. S4), we hypothesize this could be a result of: (1) lack of simultaneous $C_5H_9^+$ emission flux and $O_3$ deposition flux measurements during this study, (2) the site exhibiting little variability in $O_3$ concentrations (31-42 ppb 20th-80th percentile during 2019 study) and $O_3$ deposition velocities (-0.0011-0.027 cm s$^{-1}$ 20th-80th percentile during 2018 study) (Novak et al., 2020), which makes any additional VOC flux from ozonolysis challenging to measure, and (3) the

site being coastal near an urban center, which complicates analysis of abiotic emission sources. Thus, the lack of correlation is not necessarily indicative of a lack of a marine abiotic, heterogeneous oxidation VOC source. As a result, we use laboratory experiments to assess whether $O_3$ deposition to the seawater surface and heterogeneous oxidation of the SSML can resolve a portion of this unexplained $C_5H_9^+$, and more broadly, $C_xH_y^+$ and $C_xH_yO_z+$ emission flux of organic material.

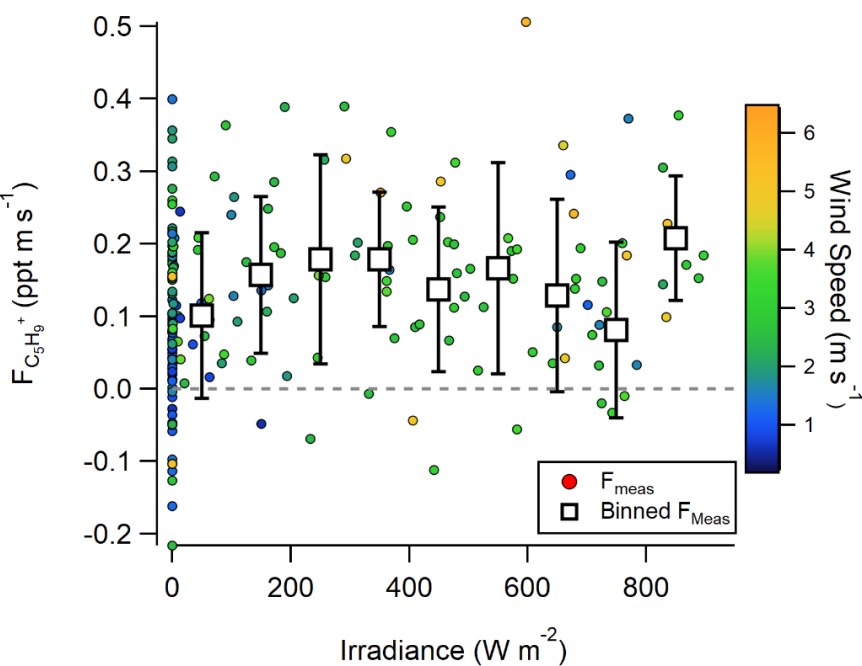




**Figure 2: Flux of C₅H₉⁺ as a function of irradiance measured at Scripps Pier in 2019. C₅H₉⁺ flux is calibrated with the isoprene calibration factor in this figure.**

## 3.2 Assessing the contribution of SSML ozonolysis to marine VOC production in laboratory experiments

A typical experiment followed the sequence presented in Table S1, and for the purpose of this paper, we will focus on results from step 4 of the sequence, corresponding to ozonolysis of a water surface. Thirteen ions measured on the RT-Vocus, all with the ion formula either $C_xH_yO_z^+$ or $C_xH_y^+$, including $C_5H_9^+$, showed a prompt response to ozonolysis of the seawater surface above a minimum threshold value, as explained by and listed in Table S3. For the rest of the paper, we will prioritize discussion of the five ions with the largest ozonolysis yields and group the remaining ions with an ozonolysis response into

"Other Ions." Given that the proton transfer reaction is favorable for the majority of VOC and the expected oxygenated products of ozonolysis reactions, we take this collection of ions to represent the total VOC produced from SSML ozonolysis (Pagonis et al., 2019). Experiments were completed using both seawater and Milli-Q separately. Since Milli-Q should have no I⁻ or DOC to drive $O_3$ deposition and VOC production, it served to capture the instrument and flow tube background (Fig. S5). As a result, abiotic VOC production from $O_3$ deposition to seawater, $\Delta VOC$ and $\Delta O_3$, respectively, are defined

according to equations 1 and 2.

$$\Delta VOC = VOC_{Seawater} - VOC_{Milli-Q} \qquad (1)$$

$$\Delta O_3 = O_{3_{Seawater}} - O_{3_{Milli-Q}} \qquad (2)$$

The average $\Delta VOC$ and $\Delta O_3$ for the experiments is shown in Fig. 3, where the different ions detected correspond to the RT-

Vocus detection of molecules produced from reactions of $O_3$ with DOC constituents. Abiotic VOC production is prompt, with total VOC peaking at 2.4 ppb after 6.6 minutes of 90 ppb $O_3$ exposure, representing an 831% increase in VOC emissions from 0 minutes $O_3$ exposure. A test where $O_3$ addition bypassed the flow tube confirmed that the prompt VOC response observed in Fig. 3 was a product of $O_3$ deposition to the seawater surface rather than reactions between $O_3$ and VOC in tubing (Fig. S6). The non-zero VOC at 0 minutes suggests that the abiotic VOC studied may also be dissolved,

either from biogenic or anthropogenic sources, leading to a small residual signal from the surface degassing. After peaking, total VOC decays to within 50% of its maximum within 5 minutes, and after 60 minutes of $O_3$ exposure, a residual 0.49 ppb VOC remains. This represents an increase of 0.23 ppb from the initial VOC at 0 minutes, implying that the VOC peak at 6.6 minutes and the sustained, low VOC at 60 minutes are produced from different reactant molecules in the DOC with varying reaction rates with $O_3$ or this is on the timescale of surface renewal replenishing the SSML with reactive DOC.




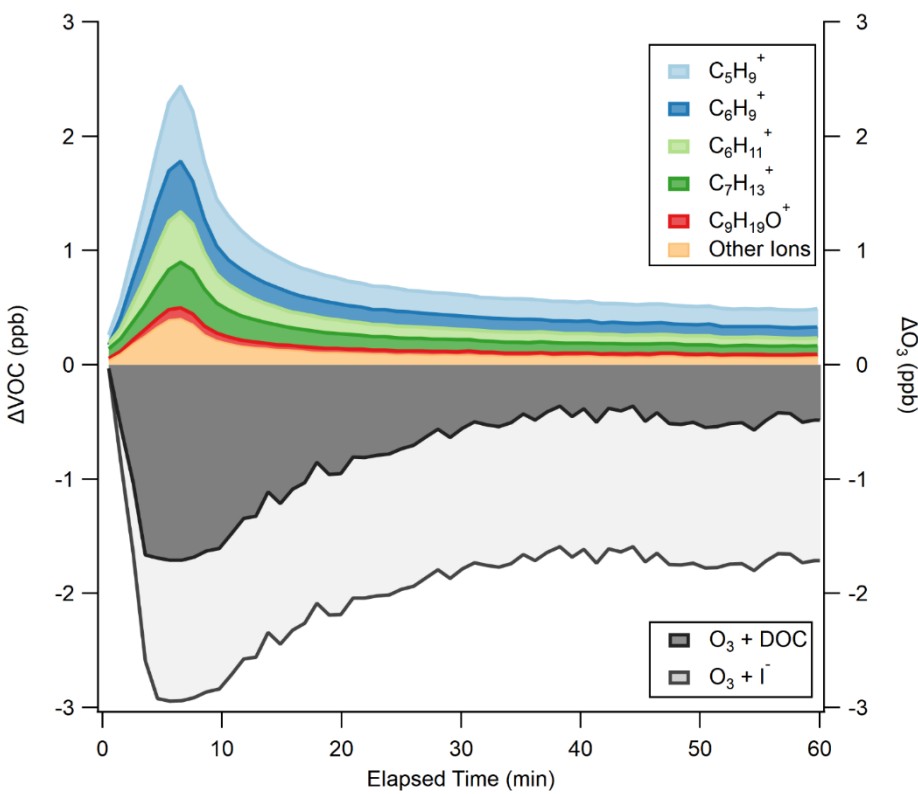

**Figure 3: ΔVOC as measured by the RT-Vocus, with individual ion contributions calibrated with an average sensitivity of 1.29 cps ppt⁻¹ stacked to sum to the total. ΔO₃ as measured by the Ox-CIMS is partitioned into calculated contributions from I⁻ (light gray) and DOC (dark gray), using the measured [I⁻] of 110.9 nM and an O₃ + I⁻ rate constant determined experimentally (Fig. S8).**

$O_3$ deposition closely follows VOC production ($R^2$ = 0.69, Fig. S7), with the largest $O_3$ loss to the surface (2.9 ppb) occurring at 6.6 minutes of $O_3$ exposure, in line with peak VOC production. $\Delta O_3$ changes quickly during the first 25 minutes of $O_3$ exposure and then stabilizes around -1.7 ppb for the remaining experiment time. The measured $\Delta O_3$ by the Ox-CIMS is a total $O_3$ loss to the surface, driven by both I⁻ and DOC. We partition the measured $\Delta O_3$ to I⁻ using the measured I⁻ concentration of 110.9 nM, which is on the higher side of reported [I⁻] but still typical for coastal regions (Chance et al., 2019), and our experimentally measured loss rate for the $O_3$ + I⁻ reaction (Fig. S8). For this calculation, we assume that surface I⁻ is constant throughout and not depleted by $O_3$, which could happen at high $O_3$ concentrations (Schneider et al., 2020). If we assume the remaining measured $\Delta O_3$ is lost to the $O_3$ + DOC reaction, I⁻ contributes 60% and DOC 40% of $O_3$ loss during the full experimental duration in Fig. 3. Using $[I^-] = 110.9\ nM$ and $k_{I^-} = 2.4 \times 10^9\ M^{-1}s^{-1}$, the calculated partitioning of $O_3$ loss corresponds to an I⁻ reactivity of 266 s⁻¹ and DOC reactivity of 177 s⁻¹. The I⁻ reactivity is within a typical range for oceanic conditions (100-300 s⁻¹) (Chance et al., 2019; Magi et al., 1997). DOC reactivity is an order of magnitude lower than what was experimentally determined in Shaw and Carpenter (2013). However, the DOC reactivity in



Shaw and Carpenter (2013) is caveated as being anomalously high, as they suspect their DOC extract used for analysis was biased toward the more $O_3$-reactive fraction and thus their rate constant might not be representative of true marine DOC.

Lastly, we use the partitioned $O_3$ reactivity to DOC and a measured $[DOC] = 32\ \mu M$ to calculate a $k_{DOC} = 5.55 \times 10^6\ M^{-1}s^{-1}$ for our seawater samples. We note that this reactivity and rate constant are derived from a bulk DOC concentration, despite the reaction likely happening at the surface. Freezing the seawater samples was required for sample preservation, but it can decrease the concentration of DOC (Peacock et al., 2015); this likely affected our samples as the DOC measurement is low compared to previous measurements, especially for coastal, surface seawater (Dafner and Wangersky, 2002). The calculated $k_{DOC}$ in our study should be taken as a lower limit and is estimated to be up to a factor of three low based on past measurements of DOC in similar regions being 70-100 µM, implying an upper limit for our study of $k_{DOC} = 1.67 \times 10^7\ M^{-1}s^{-1}$. The range of $k_{DOC}$ from our experiments is still within the range of rate constants of molecules used to approximate DOC reactivity ($k_{ethene} = 1.8 \times 10^5 M^{-1}s^{-1}$, $k_{DMS} = 8.6 \times 10^8 M^{-1}s^{-1}$, $k_{Chl-a} = 6 \times 10^7 M^{-1}s^{-1}$) (Dowideit and Von Sonntag, 1998; Gershenzon et al., 2001; Clifford et al., 2008). It is at least a factor of two lower than the existing measurement of $k_{DOC}$ for an authentic, marine sample, which the authors again suspect could be high based on the prepared extract. This comparison implies that either the DOC in our samples could be more rich in organics with slow bimolecular reaction rate constants with $O_3$, like alkenes and alkanes, than the DOC for which that rate constant was measured, or the longer sample storage in our study lowered the $k_{DOC}$, making these values less comparable (Shaw and Carpenter, 2013). Nonetheless, the lower limit ($k_{DOC} = 5.55 \times 10^6\ M^{-1}s^{-1}$) and upper limit ($k_{DOC} = 1.67 \times 10^7\ M^{-1}s^{-1}$) rate constants determined in this work provide useful constraints for marine DOC.

### 3.3 Comparison of laboratory and field yields of VOC and $O_3$ deposition

Table 1 presents total VOC and ion-specific VOC yields ($\varphi_{VOC}$) for lab and field experiments. Molecules which can contribute to the individual ions on the RT-Vocus were determined through GC-Vocus measurements of laboratory ozonolysis experiments and are presented alongside yields in Table 1. GC-Vocus results will be discussed in Sect. 3.4.

VOC yields from laboratory experiments ($\varphi_{VOC,lab}$), speciated by ions contributing to the total signal, were calculated from Fig. 3 according to Eq. (3), where areas refer to the integrated area under the curve for the $\Delta VOC$ and $\Delta O_3$ time series. VOC were quantified based on the average aldehyde calibration factor and the range in yield reflects standard deviations in $O_3$.

$$\varphi_{VOC,lab} = \frac{Area\ \Delta VOC}{Area\ \Delta O_3} \tag{3}$$

In order to compare $\varphi_{VOC,lab}$ with VOC yields from the field Scripps Pier flux measurements, where we do not have DOC measurements to be able to partition $O_3$ loss to organics, $\Delta O_3$ is taken as the total measured $O_3$ loss to both DOC and $I^-$ in



Fig. 3. Areas were calculated by integrating from 0 to 25 minutes, marked by the time point when $\Delta O_3$ was within 15% of steady-state $\Delta O_3$, to capture the VOC production from prompt ozonolysis. This resulted in an average $\varphi_{VOC,lab}$ of 0.51 (0.43-0.62). If we instead integrate the full 0 to 60 minutes of the experiment, $\varphi_{VOC,lab}$ decreases to an average of 0.41 (0.34-0.52). However, the distribution of RT-Vocus ions contributing to the total VOC yield is very similar between prompt (0-25 minutes) and steady-state (25-60 minutes) ozonolysis (Fig. S9). The decrease in yield, but largely unchanged composition of RT-Vocus ions, implies the surface concentration of reactive organics is being depleted over time and is not being replenished on the timescale of $O_3$ deposition.

VOC yields from field flux measurements at Scripps Pier ($\varphi_{VOC,field}$) were calculated according to Eq. (4).

$$\varphi_{VOC,field} = \frac{F_{VOC}}{F_{O_3}} \tag{4}$$

For the purpose of comparison, we investigate the subset of RT-Vocus ions with statistically significant responses during lab ozonolysis experiments (Table 1, Table S3). $F_{VOC}$ is the mean VOC flux measured at Scripps Pier in 2019 using the isoprene calibration factor for $C_5H_9^+$ and an average aldehyde calibration factor of 4.1 cps ppt$^{-1}$ for other VOC. $F_{O3}$ is calculated from the mean measured $O_3$ deposition velocity ($v_d$) at Scripps Pier in 2018 (0.013 cm s$^{-1}$) (Novak et al., 2020) and the mean measured $O_3$ mixing ratios during the VOC flux study in 2019 (Novak et al., 2022), according to Eq. (5). The range in $\varphi_{VOC,field}$ reported below is based on standard deviation of $O_3$ mixing ratios measured in 2019. However, we note that the uncertainty in calibration factors (Sect. S1) and VOC fluxes would drive a much larger uncertainty than just $O_3$ variability. The calculated total $\varphi_{VOC,field}$ represents an upper limit for a field VOC yield from $O_3$ deposition; without GC measurements for the field data, we cannot definitively rule out biogenic molecules detected at these RT-Vocus ions, and the calculation assumes all VOC flux is abiotic.

$$F_{O_3} = v_{d,O_3} * [O_3] \tag{5}$$

| $\varphi_{VOC,lab}$ (%) | $\varphi_{VOC,field}$ (%) | Ion | Potential Contributing Molecules |
|---|---|---|---|
| 14.1 (11.8-17.3) | 2.62 (2.22-3.20) | $C_5H_9^+$ | Pentanal, heptanal, octanal, nonanal/unidentified $C_9H_{18}O$, decanal |
| 9.19 (7.73-11.3) | 0.88 (0.75-1.07) | $C_6H_9+$ | Hexanal, nonanal/unidentified $C_9H_{18}O$ |
| 8.66 (7.29-10.7) | 0.37 (0.32-0.46) | $C_6H_{11}^+$ | Hexanal, nonanal/unidentified $C_9H_{18}O$, decanal |
| 8.66 (7.29-10.7) | 0.26 (0.22-0.32) | $C_7H_{13}^+$ | Heptanal, decanal, unidentified $C_{11}H_{22}O$ |
| 2.19 (1.84-2.70) | 0.037 (0.031-0.045) | $C_9H_{19}O^+$ | Nonanal/unidentified $C_9H_{18}O$ |



| 7.77 (6.54-9.56) | 0.51 (0.43-0.62) | Other Ions | Pentanal, hexanal, heptanal, octanal, nonanal/unidentified $C_9H_{18}O$ |
|---|---|---|---|
| **50.5 (42.5-62.2)** | **4.68 (3.97-5.72)** | **Total** | **Pentanal, Hexanal, Heptanal, Octanal, Nonanal, Decanal, Unidentified $C_9H_{18}O$ and $C_{11}H_{22}O$** |

**Table 1: Laboratory and field VOC yields from ozonolysis using measurements from the RT-Vocus and Ox-CIMS. Molecules that can contribute to ions with ozonolysis responses are provided by the GC-Vocus.**


We find that laboratory measurements of the VOC yield from seawater ozonolysis are a factor of 10 larger than those estimated from eddy covariance field measurements. There are several reasons why we might expect the laboratory measurements to overpredict the actual VOC yield: (1) Laboratory experiments were conducted using quiescent seawater with an established SSML, allowing for organic material to concentrate at the surface, thus enhancing surface heterogeneous

reactions that can produce VOC (Donaldson and Vaida, 2006; Wurl et al., 2011). In contrast, the SSML at the ocean surface is continually disrupted, reducing the likelihood that $O_3$ can react with a stagnant, concentrated organic surface (Wurl et al., 2011). (2) Ambient photochemistry is inactive in laboratory experiments, reducing the possibility for photochemical DOC transformations in the flow tube and a seawater sink of VOC due to photochemical reactions. In the ocean, produced VOC might be lost photochemically in the seawater before emission, resulting in a measured field yield lower than the comparable

lab yield (Chiu et al., 2017). (3) $O_3$ concentrations in lab experiments were roughly twice as high as in the field, allowing for the possibility that we observed reaction products in the lab that are only facilitated at high $O_3$. Additionally, any $O_3$ source variability was not directly measured simultaneously during experiments, meaning that quicker $O_3$ fluctuations than what occur in the ambient could have heightened laboratory yields. (4) Assumptions were made for the intention of lab and field comparisons that may contribute to the divergence between these measurements, including: We compare lab data to field

data that has a factor of three lower $[I^-]$ and unknown [DOC]. If we were to subtract the fraction of $O_3$ lost to the extra $I^-$ in lab experiments, the VOC yield would be even larger, making the results diverge further. Closing $\varphi_{VOC,lab}$ and $\varphi_{VOC,field}$ would require $\varphi_{VOC,field}$ to be higher, which is possible if the $O_3$ deposition flux during the 2019 field study was smaller than the value used based on $O_3$ flux measurements at this site in 2018, potentially due to lower $[I^-]$ and [DOC] in 2019 than in 2018. The reasons for the disagreement between total $\varphi_{VOC,lab}$ and $\varphi_{VOC,field}$ highlight unique challenges to doing these

experiments and caution our ability to scale laboratory-derived ozonolysis yields to the field. Future work would be improved by doing lab and field experiments at the same time, using the same water with equivalent [DOC] and $[I^-]$.

While $\varphi_{VOC,lab}$ and $\varphi_{VOC,field}$ differ in total magnitude, we show that the distribution of measured ions on the RT-Vocus share a similar trend, with $C_5H_9^+$ being the largest contributor and $C_9H_{19}O^+$ being the smallest contributor to the total VOC

yields (Fig. 4 and Fig. 5). It is worth noting that the two largest contributors to $\varphi_{VOC,field}$, $C_5H_9^+$ and $C_6H_9^+$, are two RT-Vocus ions where marine BVOC can be detected, namely isoprene at $C_5H_9^+$ and monoterpenes at $C_6H_9^+$ (Kim et al., 2010; Phillips et al., 2021). Without field GC measurements, it is possible that these numbers could be inflated by incorrectly





assigning some BVOC to an abiotic source since the ocean surface is continually being replenished and influenced by ocean biological dynamics. Additionally, some of the "Other Ions" that show the smallest ozonolysis response in lab experiments

are under the flux limit of detection in the field, meaning that the ion distribution in Fig. 4b may actually be more similar to the ion distribution in Fig. 4a as represented currently.

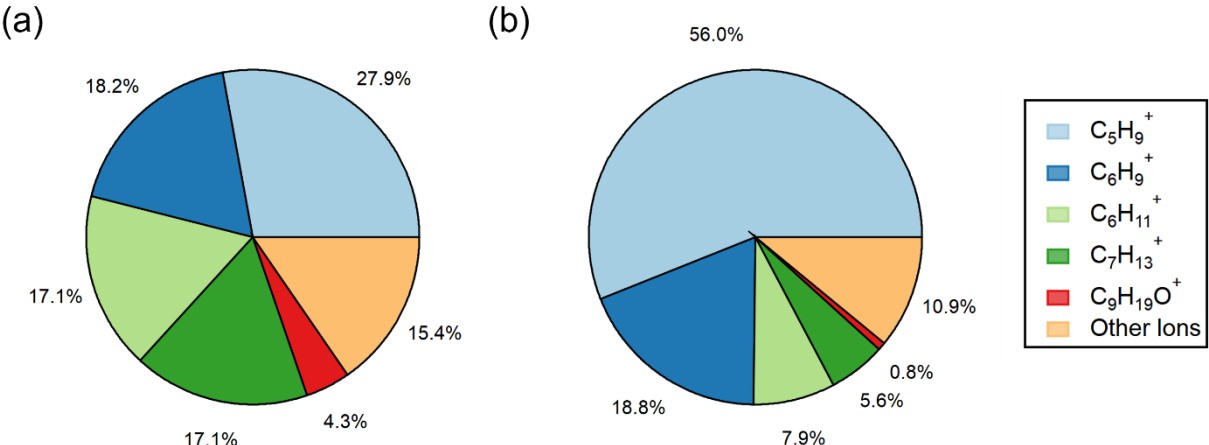

**Figure 4: Pie figure showing the ions contributing to the total VOC yield from (a) ozonolysis observed in laboratory experiments calculated using Eq. (3) and (b) ozonolysis at Scripps Pier calculated using Eq. (4). The collection of "Other Ions" refers to the sum of the ions in Table S3 that had minor laboratory ozonolysis VOC yields.**



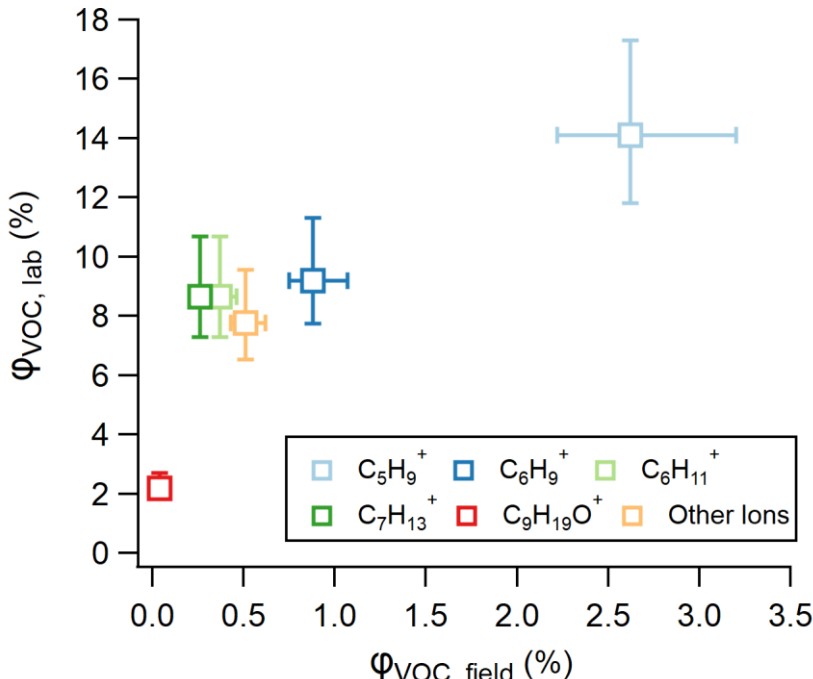

**Figure 5: Regression of VOC yields derived from laboratory and field experiments, calculated using Eq. (3) and Eq. (4),**
**respectively.**

### 3.4 Molecular Contributions to VOC yields

We use GC-Vocus measurements to qualitatively assess which molecules contribute to RT-Vocus ions. GC-Vocus measurements taken during these laboratory ozonolysis experiments indicate that RT-Vocus $C_xH_y^+$ and $C_xH_yO_z^+$ ions in Table 1 are comprised of $C_5$-$C_{11}$ OVOC, primarily aldehydes. In instances where an identified peak was not resolvable from
the elution of another molecule (ex: $C_9H_{18}O$, Fig. S10) or if a standard was not available to confirm an identification (ex: $C_{11}H_{22}O$, possibly undecanal), we report the molecular formula and designate the molecule eluting at an RT-Vocus ion as unidentified. The presence of these aldehydes in the Milli-Q ozonolysis chromatograms that were intended to serve as blanks made quantifications of the molecular contributions to each ion difficult, but we note that the larger molecules, including nonanal, decanal, and $C_{11}H_{22}O$ showed the largest and least ambiguous enhancement over background (Fig. S11). These
measurements demonstrate that RT-Vocus $C_xH_y^+$ ions in this study's conditions (E/N = 125 Td) consist of dehydrated aldehyde products (-$H_2O$) or fragments of aldehydes with a higher carbon number. To our best current knowledge, three prior laboratory studies have measured VOC products from SSML ozonolysis, namely Zhou et al. (2014), Schneider et al. (2019), and Wang et al. (2023). Despite different SSML sources and experimental conditions, our results broadly align with the previously measured distribution of molecules based on carbon number and inclusion of carbonyl moieties. One noted
difference is the lack of acetone, propanal, or acetaldehyde production in these experiments, compared to Wang et al. (2023).



Importantly, **t**his study reveals that the RT-Vocus $C_5H_9^+$ signal from ozonolysis of coastal seawater has no contribution from isoprene, but rather is a fragment of larger oxygenated VOC. A test where isoprene was added to the headspace of the flow tube containing seawater confirmed that isoprene can be measured by the GC-Vocus as configured in this study with a

detection limit of 8 ppt, calculated according to Claflin et al. (2021), implying that any abiotic isoprene or residual degassing biogenic isoprene from the seawater samples is either not present or below the detection limit (Fig. S12). This confirms that $O_3$ deposition to the seawater surface can resolve a portion of the unexplained $C_5H_9^+$ emission flux at Scripps Pier. While detection of long-chain aldehydes at $C_5H_9^+$ is not new (Ruzsanyi et al., 2013; Vermeuel et al., 2023; Wang et al., 2023), this study demonstrates fragmentation of long-chain aldehydes to $C_5H_9^+$ is an important consideration for marine atmospheric

chemistry, where $C_5H_9^+$ has long been interpreted as isoprene within the PTR-MS community (Ciuraru et al., 2015b; Pagonis et al., 2019). Our results suggest that one should proceed with caution when interpreting RT-Vocus $C_5H_9^+$ signals in regions where isoprene concentrations are low, such as in marine environments, and where aldehyde concentrations could be large, such as in urban centers (Coggon et al., 2023).

## 4 Conclusions

Results of this study show that the reactivity of $O_3$ to marine DOC can be large, comparable to $I^-$ reactivity, and can lead to the production of VOC from $O_3$ deposition to seawater. Taking $\varphi_{VOC,field}$ and $\varphi_{VOC,lab}$ as lower and upper limits, respectively, on VOC yields from ozonolysis, this study suggests that for an $O_3$ deposition rate of 0.04 ppb h$^{-1}$ (based on an $O_3$ concentration of 30 ppb, deposition velocity of 0.02 cm s$^{-1}$, and marine boundary layer height of 500 m), the instantaneous marine boundary layer VOC production rate from the surface of coastal seawater is 2-20 ppt h$^{-1}$. Given marine

biogenic DMS or isoprene mixing ratios are typically less than a couple of hundred ppt in coastal areas (Lee et al., 2004; Shaw et al., 2010), this range of VOC from ozonolysis can be a significant, unaccounted for, marine VOC source in coastal regions. For the average $O_3$ deposition flux of $1.5 \times 10^{10}$ molecules cm$^{-2}$ s$^{-1}$ ([$O_3$] = 30 ppb, $v_d$ = 0.02 cm s$^{-1}$) and scaling by an average VOC structure containing 8 carbons and a range of yields between $\varphi_{VOC,field}$ and $\varphi_{VOC,lab}$, this analysis indicates ozonolysis could source 12.6 to 136 Tg C yr$^{-1}$, competitive with the DMS source of 21.1 Tg C yr$^{-1}$. Since laboratory

experiments favor a stable SSML that is not representative of oceanic conditions, we suggest that actual yields are closer to the lower limit of this study.

Furthermore, we encourage additional work quantifying the speciated abiotic VOC composition from ozonolysis to help clarify the $C_5$-$C_{11}$ OVOC observations in this study. $C_5$-$C_{11}$ aldehydes react with OH roughly an order of magnitude faster than DMS ($k_{DMS+OH} = 4.80x10^{-12}$ and for one example, $k_{nonanal+OH} = 3.60x10^{-11}$ cm$^3$ molec$^{-1}$ s$^{-1}$) (Atkinson et al.,

2004; Bowman et al., 2003; Jimenez et al., 2007; Papagni et al., 2000), suggesting this collection of molecules could have significant influence on marine atmospheric oxidative capacity even if their emissions are at the lower limit of our study. While mechanistic aldehyde-OH oxidation has been studied in high NO$_x$ cases, to our best current knowledge, this has not

been studied under marine-relevant conditions where $NO_x$ <50 ppt (Lee et al., 2009). Similarly, much of the research
investigating SOA yields of individual primary-emitted aldehydes is completed under high $NO_x$ conditions (Chhabra et al., 2011; Chacon-Madrid et al., 2010; Chacon-Madrid and Donahue, 2011), with the only studies in the low $NO_x$ regime focused on aldehydes like pinonaldehyde that are intermediates in the oxidation of common BVOC (Chacon-Madrid et al., 2013). The long-chain acyclic aldehydes that contribute to measured RT-Vocus ions in this study have fast reaction rates with OH, are susceptible to photolysis, and are expected to form SOA based on observed new particle formation and growth
during ozonolysis of an SSML in Schneider et al. (2019). As a result, we recommend future work to investigate the oxidation and SOA yields of $C_5$-$C_{11}$ aldehydes under marine-relevant, low-$NO_x$ conditions as they have the potential to be a significant abiotic marine VOC emission source in coastal regions.

**Data availability**

Field measurements of eddy covariance VOC fluxes, $O_3$ concentrations, and irradiance, and time series of laboratory
seawater ozonolysis experiments are available at http://digital.library.wisc.edu/1793/84597 (Kilgour et al., 2023).

**Supplement**

Additional discussion of calibration factors, uncertainties, calculation of dissolved isoprene concentrations, and supporting figures and tables.

**Author contributions**

DBK, GAN, THB designed the research. DBK conducted laboratory experiments and analyzed the data. GAN conducted the field study and analyzed the data. MC and BL work for Aerodyne Research, Inc., which developed the GC-Vocus instrument used in this study and provided instrumentation support and feedback. DBK and THB wrote the paper. All authors edited and reviewed the paper.

**Competing interests**

The authors declare that they have no conflict of interest.

**Acknowledgments**

The authors gratefully acknowledge the contributions of Steve Myers and Blaise Thompson at University of Wisconsin-Madison Chemistry in building the flow tube used for the laboratory experiments. The authors thank the staff at Scripps Pier,



Scripps Institution of Oceanography for their support of the flux study, Alexia Moore at University of California, San Diego

for seawater collection, Christopher Jernigan at University of Wisconsin-Madison for help with the Ox-CIMS, Neal Arakawa with the Environmental and Complex Analysis Laboratory at University of California, San Diego for measuring iodide in the seawater samples, and James Lazarcik at the University of Wisconsin-Madison Water Science and Engineering Laboratory for support with measuring DOC in seawater samples.

**Financial support**

This work was supported by the National Science Foundation (AGS 1829667) and the National Science Foundation Center for Chemical Innovation Center for Aerosol Impacts on Chemistry of the Environment (CHE 1801971).

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
