# Peer review of "Production of oxygenated volatile organic compounds from the ozonolysis of coastal seawater"

_EGUsphere, 2023_

## Author Comment (AC1)

Comments for all Reviewers: Thank you for reviewing this manuscript. We appreciate the insightful comments that have helped improve the quality of the manuscript. Reviewer comments are reproduced below in blue text. Author responses are in black text. For edits, additions to the text are in red, and deletions are represented with a red strikethrough. Line numbers referenced in the author responses reflect the line numbers in the expanded, tracked changes manuscript and SI.

Reviewer 1:

This paper by Kilgour et al. explores the emission of volatile organic compounds (VOCs) from the ozonolysis of seawater with a combination of field and laboratory measurements. From this work, the authors determine the approximate yield of VOCs from ozone deposition, which is about a factor of 10 larger in the lab experiments than at Scripps Pier during field measurements. Using a proton transfer reaction mass spectrometer, the authors also tentatively identify the emitted VOCs as being primarily aldehydes, although there are limitations to their analytical method.

This work shows the yield of VOCs is competitive with DMS, which suggests that it could be an important source to consider when thinking about secondary aerosol formation or evaluating the oxidative capacity of the marine atmosphere.

General comments:

I am very interested to know more about the difference between the laboratory and field measurements. The authors provide 4 potential reasons for why the laboratory measurements had a higher ozone yield for VOCs relative to the field measurements, however some ideas could be discussed more thoroughly.

- The authors first reason for lower field yields is the relatively stable SSML in the lab relative to the more dynamic real SSML in the ocean. Did they authors specifically sample the SSML at Scripps? My impression is they collected the underlying seawater, which contained some unsaturated/insoluble species which then formed a new microlayer in the lab. Do the authors think that the composition between the laboratory SSML and the authentic SSML composition could impact the VOC yield? Two factors come to mind, including the composition of the SSML (i.e. the reactive component) and changes to the physical partitioning of the VOCs from the aqueous phase into the gas-phase.

Thank you for bringing this up. We did not specifically sample the SSML at Scripps. As you said, we collected the near-surface seawater which then formed a new microlayer in lab. It is possible that the composition of the SSML reactive component between the field and lab samples had changed, as freezing and storage can decrease the concentration of DOC and preferentially degrade different DOC moieties (Peacock et al., 2015; Schneider-Zapp et al., 2013). Changes to the physical partitioning of VOC could also occur as the field samples have increased turbulence and disruption of the SSML (discussed in Reason 1, L397). Partitioning dissolved VOC from the aqueous phase to the gas phase results in larger emission fluxes at higher turbulence for nonsurface reactions. However, since this multiphase reaction is a surface reaction requiring an SSML to establish, we think that the continuous disruption of the SSML played a larger role in the divergence of our results than any changes in the physical partitioning term, which we comment on in Reason 1 in the text.

- The second reason the authors hypothesize difference is the presence of photochemistry, which would be occurring in the environment and could reduce the lifetime of the VOCs in seawater to lower the flux. (L373) From Figure 1, the emission of VOCs has a diurnal profile which peaks in the afternoon, which is likely related to the ozone flux. Can the authors show the VOC yield from ozonolysis changes during the daytime and nighttime during their field study? This would perhaps disprove their second point.

Thanks for this point. We have looked into whether the VOC yield from ozonolysis changes throughout the day during the field study in the figure below. We have done this following Eq. 4 for the same set of ions in the MS, using time-matched VOC flux measurements and $O_3$ flux measurements based on measured $[O_3]$ and the average $v_{d,O3}$ from 2018. We cannot draw a conclusion here as the morning period is limited by few measurements because winds were often from land during this time. Roughly 30% of data is in the nighttime (Hr. 20-8) whereas 70% of data is in the daytime (Hr. 8-20). As a result, we do not speculate on this further in the main text.

[Figure]

- Is there any evidence towards biological factors reducing the lifetime of VOCs in the aqueous phase? For example, biological processes have been shown to be a sink of acetone (https://www.ncbi.nlm.nih.gov/pmc/articles/PMC7983863/ ), and I assume other VOCs as well based on recent reviews and studies, (https://www.sciencedirect.com/science/article/pii/S0012825223000491 , https://www.nature.com/articles/s41564-020-00859-8 )

This is an interesting point, thank you for pointing out some references for this. We agree that biological sinks of these molecules might be present and could differ in the lab vs. field experiments due to seawater vertical gradients in the field and sample storage conditions. We speculate on this in the combined response to the next suggestion.

- The role of downward mixing in the ocean vs the lab. In the iodide-ozone model built by Carpenter et al. (2013; https://www.nature.com/articles/ngeo1687), they found the rate of emission also depends on the downward mixing of the oxidized iodine, which sequesters the volatile products in the bulk ocean. This seems likely to be occurring with the organics, as well and should be proposed as a limitation of the laboratory measurement that could contribute to differences in yield.

Thank you for bringing this up. The SSML is continually replenished in the ocean, but it is possible that organic ozonolysis products could get vertically mixed downward in the ambient and sequestered in the ocean. In the laboratory experiments, ozonolysis is occurring on a quiescent surface where vertical mixing is limited. This is a limitation of the laboratory experiments in that they are not fully reproducing the turbulence observed in the ocean. By not reproducing this potential sequestration of organics in the bulk ocean, it contributes to the laboratory yield being higher than the field yield.

L397: "Because of this turbulence at the ambient ocean surface, it is possible that any oxidized products could also get vertically mixed downward and sequestered in the bulk ocean, which would further reduce the field yield relative to the laboratory yield. Similarly, the ambient ocean has vertical gradients in biological processes that could act as sinks for VOC products and contribute to the reduced field yield (Halsey and Giovannoni, 2023)."

The authors also make an important point that under marine conditions, the $C_5H_9^+$ ion cannot be interpreted as being isoprene, since it is a fragment of many different, larger aldehydes. The authors used a GC-Vocus system to resolve the parent ions, thus separating the different soures of the $C_5H_9^+$ ion fragment. Does the results from these experiments match the frgment library put together by Pagonis et al. (2019)? Why or why not? What parent ion contributes the most (if any) to the total $C_5H_9^+$ ion signal?

The fragment library put together by Pagonis et al. (2019) reports the collection of m/z at which an individual molecule can be detected in PTR-MS and notes the percentage of the signal going to each m/z, determined either by calibration or by when a GC-MS pre-separation method was used. We have used this data to determine the ratio of the ion where isoprene is detected at m/z 69.070 to the parent ion, calculated as the ratio of their signal percentages. We compare this value to the ratio of the fitted m/z 69.070 ion peak to the parent ion peak at the retention time of the molecule on the GC-Vocus during experiments where we ran standards of these molecules. A value greater than one indicates that more signal is going to the m/z 69.070 $C_5H_9^+$ fragment where isoprene is detected compared to the parent m/z. We do this for the collection of aldehydes which we have standards for that were also in the fragmentation library in Pagonis et al. (2019) at the time of this publication.

We find that within the existing PTR-MS library for these aldehydes, the contribution of m/z 69.070 to the aldehyde signal is not consistent, with several examples not reporting any

contribution of m/z 69.070. It is expected that at higher E/N, greater fragmentation exists, and the ratio of m/z 69.070 to the parent ion would be larger. Without each reference providing an E/N condition for their results, it is difficult to determine why the inconsistencies exist.

Our work qualitatively agrees with the library that m/z 69.070 is a significant fragment of these aldehydes, but reports an even larger contribution of the $C_5H_9^+$ fragment. This is potentially due to the higher E/N in this study (E/N = 125) compared to what is reported in the database and any potential background of these aldehydes in our GC-Vocus system described in the main text. Our work shows that $C_5H_9^+$ was a significant fragment for all four of these aldehydes, and was particularly dominant for pentanal, octanal, and nonanal.

We have included this in the text:

L459: "The detection of long-chain aldehydes at $C_5H_9^+$ is not new (Ruzsanyi et al., 2013; Vermeuel et al., 2023; Wang et al., 2023), and the findings of this work are qualitatively consistent with those in the fragmentation library from Pagonis et al. (2019), where $C_5H_9^+$ made large contributions to the total detected signals of pentanal, octanal, and nonanal in some PTR-MS studies."

| Molecule | Reference | E/N Ref. | % m/z 69.070 in reference | % m/z parent in reference | Ratio m/z 69.070 / parent ion in reference | Ratio m/z 69.070 / parent ion in this work |
|---|---|---|---|---|---|---|
| Pentanal | Buhr IJMS 221, 1, 2002 | Not stated | 75.0 | 5.0 | 15 | 16.5 |
|  | Spanel, IJMSIP 165/166, 25, 1997 | 0 | 25.0 | 75.0 | 0.33 | 16.5 |
|  | Warneke, ES&T 37, 2494, 2003 | 106 | 78.0 | 22.0 | 3.5 | 16.5 |
| Heptanal | Buhr IJMS 221, 1, 2002 | Not stated | 4.0 | 5.0 | 0.8 | 3.7 |
|  | Koss, ACP 18, 3299, 2018 | 120 | 0 | 100 | 0 | 3.7 |
|  | Spanel, IJMS 213, 163, 2003 | 0 | 0 | 80.0 | 0 | 3.7 |
|  | Warneke, ES&T 37, 2494, 2003 | 106 | 0 | 48.0 | 0 | 3.7 |

| | | | | | | |
|---|---|---|---|---|---|---|
| Octanal | Buhr IJMS 221, 1, 2002 | Not stated | 42.0 | 11.0 | 3.8 | 17.5 |
| | Spanel, IJMS 213, 163, 2003 | 0 | 0 | 85.0 | 0 | 17.5 |
| | Warneke, ES&T 37, 2494, 2003 | 106 | 0 | 68.0 | 0 | 17.5 |
| Nonanal | Buhr IJMS 221, 1, 2002 | Not stated | 37.0 | 19.0 | 1.9 | 7.9 |

Specific comments:

L75 – Authors state that the flux of carbon from oxidation is "competitive with the carbon mass flux from BVOC and a proposed photochemical source". The authors provide an estimated range from a previous study for oxidation, but not for photochemical or BVOC flux; perhaps it would be clearer if all the ranges (and the limitations of their estimates, perhaps) were presented.

Thank you, these numbers are BVOC flux are in lines 34-35 and from photochemistry in line 51. We agree that restating them here with a small discussion of their limitations and uncertainties helps with clarity. We have also updated the estimated flux from DMS with a newer climatology, Hulswar et al. (2022), as suggested by Reviewer 2.

The new text reads:

L75: "Using an average $O_3$ deposition flux ($1.5 \times 10^{10}$ molecules cm$^{-2}$ s$^{-1}$ corresponding to an $O_3$ concentration of 30 ppb and deposition velocity of 0.02 cm s$^{-1}$), Novak and Bertram (2020) estimated that the carbon mass flux of VOC from ozonolysis of the seawater surface to be 17.5-87.3 Tg C yr$^{-1}$ (for $\varphi_{VOC}$ ranging 0.1-0.5) (Novak and Bertram, 2020), competitive with the carbon mass flux from BVOC (e.g. DMS flux estimated at 20.3 Tg C yr$^{-1}$) (Hulswar et al., 2022) and a proposed photochemical source (23.2-91.9 Tg C yr$^{-1}$) (Brüggemann et al., 2018). It is important to note that each estimate represents an average over large spatiotemporal variability and comes with limitations and uncertainties. For example, the BVOC DMS estimate is based on the dataset of dissolved DMS concentrations available, the non-DMS BVOC estimate is not well-constrained due to limited measurements, and the proposed photochemical and ozonolysis terms are based on meteorological estimates and scaling of laboratory yields.  Nonetheless,"

L235 – The authors state that degassing experiments observed BVOC like DMS, and reference Figure S3 which shows the emission of DMS and isoprene. Are these the only two BVOC ions observed? How were these attributed to BVOCs and not photochemical sources or other sources?

The intention of this statement is to point out that in conditions where we measured VOC degassing, we did not observe isoprene. This chromatogram was taken during Step 3 of the experiment in Table S1, where we are measuring molecules degassing from seawater in a zero air overflow. The degassing molecules could have anthropogenic sources (ie. runoff) or have biological sources. We expect photochemistry to have a negligible effect on this portion of the experiment, as indoor light sources produce wavelengths too long to majorly affect photochemistry and the flow tube's distance from the ceiling lights was large. While DMS is not the only BVOC ion observed, it is well-documented to have biological sources, so we use it to show that when we measure BVOC, we do not measure isoprene. Other ions observed in this portion of the experiment could have more complicated degassing sources (anthropogenic and biological), so we keep the comparison to DMS since it is so well-studied. Reporting additional BVOC is more speculative and outside of the scope of this paper.

L263 – How was this threshold value (of 50 cps) chosen?

The threshold of at least 50 cps at the peak was chosen to ensure that data analysis and interpretation over the entire experiment was above a conservative instrument detection limit. Based on the thresholds described to determine ions with a prompt ozonolysis peak, then the initial signal was at least a factor of two lower and the final signal was at least a factor of 1.5 lower. 50 cps at the peak provided a threshold to see the entire ion time series.

We have included this:

SI L143: "The signal threshold was chosen to ensure that data analysis over the entire experiment was above a conservative instrument detection limit."

L439 – Why 8 carbons? Previously, Novak & Bertram (2021) used 5 carbons.

In Table 1, we show the potential contributing molecules to RT-Vocus ions that showed an ozonolysis response. These include C5-C11 molecules, where 8 carbons is in the middle and 5 carbons is the lower limit. As such, we chose to use 8 carbons to estimate the reactive carbon source from ozonolysis since it is more in line with the experimental findings from this work.

This has been added:

L476: "average VOC structure containing 8 carbons (based on the median carbon number of molecules in these experiments) and a range of yields …"

Errata

L270 – different color?

Thanks for the catch, we have made sure all text is black in the updated version.

Reviewer 2:

This manuscript presents a unique combination of field observations of reactive organic carbon (ROC) fluxes and laboratory experiments probing reactive organic carbon emissions from a quiescent seawater microlayer reacting with gas phase ozone. The authors provide VOC yield estimates from both in situ ROC fluxes and laboratory ozone oxidation, and demonstrate that real world VOC yields are likely much smaller than laboratory estimations suggests. This is a very important point to make given that laboratory VOC yields from similar experiments have already been scaled to global emissions in some modelling studies. However, even scaling the much lower field VOC yield by an average ozone deposition flux points to an important source of ROC over the world's oceans. This work also makes two additional very useful points: (1) that the authors see no evidence for direct photochemistry driving VOC fluxes from the sea surface, and (2) that C5H9+ signals from PTR-type instruments should be interpreted with extreme caution in marine environments (i.e., building on Coggon et al., 2023 for urban centers).

General Comments

1. (L19, L444) 21.1 Tg-C/yr from marine DMS emissions. It is difficult to understand where this number comes from. It is not cited, and doesn't appear to line up with current climatologies. For example, the third revision of the DMS climatology (Hulswar et al., 2022 https://essd.copernicus.org/articles/14/2963/2022/) estimates a global DMS emission of ~27.1 Tg-S/yr. Scaling to carbon in DMS gives ~20.3 Tg-C/yr. While this is relatively close to the number stated in the abstract and conclusions, this highlights the ambiguity in the origin for this comparison. The authors should cite recent climatologies for this comparison and make clear that this number is both not exactly known and arises from averaging over significant regional variability. Otherwise, this may not be a useful benchmark against which to judge the importance of VOC emissions driven by O3 deposition to the sea surface.

Thanks for the comment, our number was based on the Lana et al. (2011) climatology but we mistakenly did not cite it throughout. Thank you for bringing up that newer climatologies exist. We have used the Hulswar et al. (2022) number throughout and cited it. We have also included a comment where we compare emission terms from different VOC production pathways that all have some degree of uncertainty. Text is as follows:

L19: "competitive with the DMS source of approximately 20.3 Tg C yr$^{-1}$"

L34: "Marine DMS emissions are estimated to be roughly 20.3 Tg C yr$^{-1}$ (Hulswar et al., 2022)"

L78: "competitive with the carbon mass flux from BVOC (e.g. DMS flux estimated to be 20.3 Tg C yr$^{-1}$) (Hulswar et al., 2022) … It is important to note that each estimate represents an average over large spatiotemporal variability and comes with limitations and uncertainties. For example, the BVOC DMS estimate is based on the dataset of dissolved DMS concentrations available, the non-DMS BVOC estimate is not well-constrained due to limited measurements, and the proposed photochemical and ozonolysis terms are based on meteorological estimates and scaling of laboratory yields."

L478: "competitive with the DMS source  estimated at 20.3 Tg C yr$^{-1}$ (Hulswar et al., 2022)"

2. Use of the term heterogeneous (L29, L369 and elsewhere). Given the reactions studied occur at liquid interfaces, "multiphase" may be more appropriate. e.g., see https://acp.copernicus.org/articles/23/9765/2023/ for current suggestions on revisions to terminology. It is more useful to consider a multiphase reaction that may have substantial interfacial and bulk components, then it is to consider a perfectly "heterogeneous" surface-only reaction versus and entirely bulk phase reaction.

Thanks for pointing this out and we appreciate the paper reference for more clarity. We have replaced the term heterogeneous throughout the text with multiphase based on Abbatt and Ravishankara (2023).

3. The estimation of multiphase O3 reactivity and k_DOC (L299 - L324) requires further, and more nuanced, discussion.

(a) The authors apportion O3 reactivity into an I- reactivity of 266 s^-1 and a DOC reactivity of 177 s^-1. Can the authors propagate uncertainties in the measurements that go into this estimation of O3 reactivity? Further, this O3 reactivity assumes a bulk rate coefficient and a bulk iodide concentration, and would be more accurately referred to as a "bulk reactivity," because (as the authors state later in the paper) the O3 reactions likely have a significant interfacial component.

The reported $I^-$ reactivity is the product of existing, literature values for the $I^- + O_3$ rate constant ($k_{iodide}$) and the measured concentration of iodide in the samples. We have updated the value of $k_{iodide}$ based on your suggestion of Shaw and Carpenter (2013), which is $1.4 \pm 0.2 \times 10^9$ M$^{-1}$ s$^{-1}$ at 20 °C and pH 8. The concentration of $I^-$ (110.9 nM) was only measured once. However, the samples were all collected at the same time and from the same location, so we expect the variability in the $I^-$ concentration is small and the uncertainty is from the measurement technique. IC-ICP-MS is sensitive to iodide with detection limits much lower than what was observed in the sample (e.g. 0.80 nM in Shi and Adams, 2009). As a result, we set the uncertainty in $I^-$ reactivity as 20% based on 14% from the rate constant and an estimated ~5% from the iodide concentration measurement. We note that the $k_{iodide}$ rate constant varies by larger than 20% in the literature, and the absolute magnitude of our $I^-$ reactivity is ultimately dependent on this.

To apportion the fraction of ozone that reacts with iodide (here 60%) and with DOC (here 40%), we compare our measurements of ozone loss to seawater with our measurements of ozone loss to Milli-Q water with equivalent iodide concentrations (as shown in Figure S8). The slope of that line was used to apportion $O_3$. As a result, the uncertainty in this apportionment of $O_3$ reactivity is a function primarily of the $O_3$ measurement (error bars on Figure S8). Based on the range in [$O_3$] in Fig. S8, this could more accurately be reported as 40% ± 14% (26-54%) of $O_3$ reactivity is due to reactions of ozone with DOC.

The absolute magnitude of the DOC reactivity (in units of s$^{-1}$) is calculated from the $I^-$ reactivity (a function of [$I^-$] and the rate constant), and the fraction of $O_3$ that reacts with DOC (here 40%).

As a result, the uncertainty in DOC reactivity is from the uncertainty in the $O_3$ measurement and the $I^-$ reactivity.

We have also updated the language to bulk reactivity.

L317: "The uncertainty in the apportionment of $O_3$ reactivity between $I^-$ and DOC is dependent on the measurement of $O_3$ loss to $I^-$ only solutions (at equivalent $[I^-]$ found in the seawater samples) as presented in Fig. S8. Using $[I^-] = 110.9\ nM$ and $k_{I^-} = 1.4 \pm 0.2 \times 10^9\ M^{-1}s^{-1}$, the calculated partitioning of $O_3$ loss corresponds to a bulk $I^-$ reactivity of 155 s$^{-1}$ $\pm$ 20% and a bulk DOC reactivity of 104 s$^{-1}$ $\pm$ 35%. The uncertainty in the absolute magnitude of $I^-$ reactivity is propagated from uncertainty in the chosen bulk $k_{I^-}$ rate constant ($1.4 \pm 0.2 \times 10^9\ M^{-1}s^{-1}$) (14%) (Shaw and Carpenter, 2013) and measurement of $I^-$ concentration (5%). The uncertainty in the absolute magnitude of DOC reactivity is propagated from the $I^-$ reactivity (~20%) and uncertainty in the fraction of $O_3$ that reacts with DOC (14%), giving a total uncertainty of ~35%.

(b) In addition to assuming the surface I- is not depleted by reaction, do the authors also assume that the surface I- concentration is equal to the bulk I- concentration? I- ions are known to have affinity for the air-water interface, and a Langmuir adsorption isotherm, with literature constraints on the bulk-surface partition coefficient, could be used to better constrain the magnitude of potential surface concentrations in the present experiments.

In this analysis, we assume that surface and bulk iodide concentrations are equivalent, and that surface iodide is not depleted over the course of the experiment. This is a limitation of the study, but a more detailed analysis is beyond the scope of this study, where the intention here is to provide constraints on VOC production from $O_3$ reactions. The reviewer is correct in that a Langmuir adsorption isotherm could more accurately portray the vertical distribution of $I^-$ ions showing their concentration at the interface (Moreno et al., 2018). In response to reviewers' comments, we have decided to focus solely on bulk phase reactivity, especially given that the apportionment of the total $O_3$ loss is not dependent on this distinction. That said, our determination of the $O_3$+DOC reactivity is dependent on our selection of $k_{I^-}$ and $[I^-]$ which could be a function of depth. While molecular dynamics simulations suggest that iodide is enhanced near the surface, surface tension measurements indicate that iodide is depleted over the inhomogeneous interfacial region relative to the bulk. As a result, if ozone were to react over the entirety of the interfacial region, there is less iodide here than the bulk. In contrast, if ozone were to react at the air-water interface, simulations suggest that $I^-$ may be enhanced over the bulk.

L309: "For this calculation, we assume that the near surface $[I^-]$ is equal to the measured bulk $[I^-]$ concentration, that the surface and bulk $I^- + O_3$ rate constants are equal, and that iodide is not depleted by $O_3$, which could happen at high $O_3$ concentrations (Schneider et al., 2020). While this is an over simplification of the near surface chemistry of this reaction, as discussed in (Prophet et al., 2024), a more detailed treatment of this chemistry is beyond the scope of this analysis. In the absence of chemical reaction, it is possible that the near surface iodide concentration is slightly different than the bulk iodide concentration (dos Santos et al., 2008)."

(c) Similarly, while the estimated k_DOC uses a bulk [DOC], it also uses an observed gas phase O3 loss and so inherently incorporates both interfacial and bulk O3 reactions of unknown relative

importance. It cannot be simply stated (L311) that this reaction "likely happens at the interface," rather for a bimolecular rate coefficient on the order of 5e6 M^-1 s^-1, it is possible that interfacial reactions will make a substantial contribution to O3 loss, though that will depend on the interplay between O3 and DOC partitioning to the interface (e.g., https://pubs.acs.org/doi/full/10.1021/acs.jpca.2c03059)

Thanks for bringing this up. The derived DOC reactivity and $O_3$ + DOC rate constant was determined from measurements of bulk DOC concentration, despite the possibility that the reaction may be occurring near the surface where the DOC concentration and speciation may be entirely different. It is possible that interfacial reactions could make a large contribution to $O_3$ loss, which depends on the partitioning of organic carbon at the interface (Willis and Wilson, 2022). Upon further review, we have decided to remove the discussion of $k_{DOC}$ from this paper (last paragraph of Section 3.2). The experiments in this paper were designed to investigate VOC production from $O_3$ reactivity to DOC, and were not intended to determine interfacial and bulk rate constants. As a result, we feel that the uncertainty in the $k_{DOC}$ we have retrieved from the current experiments diminish the value of adding it to this paper.

(d) The authors go on (L317) to compare their mixed bulk/interfacial k_DOC to a set of strictly bulk rate coefficients for species used to approximate DOC. These bulk, bimolecular rate coefficients span 3 orders of magnitude, and may not represent species that are relevant to marine DOC. The comparison to an "authentic marine sample" (L319) is presumably that from Shaw & Carpenter 2013 (2.6e7 M^-1 s^-1), and is therefore a more appropriate comparison as this literature value arises from similar O3 deposition experiments and is thus a combination of bulk and interfacial O3 reactivity.

We appreciate this comment, and we agree that comparing to the authentic marine sample from Shaw and Carpenter (2013) would have been most similar value to what we are reporting. As we discussed above, based on the comments we received and further thinking about it, we have decided to remove the calculation and discussion of $k_{DOC}$ in the main text. We have cut the material that this comment was referencing.

4. Figure 4: Pie charts an interesting choice here as you can't show a range or variability in these contributions. A bar chart with error bars that correspond to some measure of variability in contributions to your observed yields would be much more informative.

Thanks, we agree that pie charts don't show range or variability in contributions, but find them easier to interpret visually. However, Figure 5 addresses your comment. This figure does have error bars for the range in lab and field yields to show the measure of variability in the observations.

5. The authors use field and lab average yields to bracket the range of possible VOC yields in the conclusions and abstract. It may be more appropriate to include the range of yields for both lab and field data, show in Table 1, into these discussions.

Thank you, we agree this is a more fair interpretation of the data and we have adjusted accordingly.

L19: "results in an emission source of  10.7 to  167 Tg C yr$^{-1}$"

L478: "this analysis indicates ozonolysis could source  10.7 to  167 Tg C yr$^{-1}$"

Specific Comments

L36: "This collection of VOC" -- this wording is somewhat confusing; which VOCs specifically?

Thanks, this refers to the biogenic VOC we discussed previously. We have updated the text:

L36: " DMS, isoprene, and monoterpenes have"

L195: "Ions without an expected molecule" -- does this mean ions without a known, or single, contributing molecule?

Yes, this means ions without a known molecule contributing. We have updated the text:

L201: "Ions without  a known contributing molecule"

L302: k_iodide = 2.4e9 M^-1 s^-1, presumably this value is from Magi et al., 1997? More recent determinations exist e.g., https://pubs.acs.org/doi/10.1021/ic000919j (for pH 6.7, 1.2 (+/-0.1)e9 M^-1 s^-1). Further, given the pH dependence of O3 + I-, is the Magi 1997 rate coefficient applicable to seawater pH? This should be discussed further.

Thank you for pointing this out. Yes, we had used the $k_{iodide}$ value from Magi et al. (1997). We agree that more recent determinations fit out experimental conditions better and have updated the number (and where it is used in calculations). We have decided to use $k_{iodide}$ = 1.4E9 M$^{-1}$ s$^{-1}$ at 20 °C and pH 8 determined by Shaw and Carpenter (2013) as it was determined at low iodide concentrations and seawater pH.

This changes the numbers for DOC and iodide reactivity discussed in Section 3.2 (L319-326). We now report an iodide reactivity of 155 s$^{-1}$ and DOC reactivity of 104 s$^{-1}$. We have also cited the $k_{iodide}$ rate constant from Shaw and Carpenter (2013) in the introduction (L59) instead of that of Magi et al. (1997).

L321: Alkanes reacting with ozone?

This has been removed.

L332: "..range in yield reflects standard deviations in O3." -- +/- 1 sigma, or more?

Yes, all reported standard deviations reflect 1 sigma. We have updated the text:

L355: "range in yield reflects 1 sigma standard deviations in O$_3$"

L342-344: What fraction of organic carbon goes to the gas phase in the experiment? The authors appear to have the data to estimate this from the DOC measurement in sea water together with integrated delta-VOC over time.

We have calculated this by converting the DOC measurement for dissolved phase organic carbon to mol C cm$^{-3}$. The total VOC yield was calculated as the delta VOC measurement integrated over time to get a value in ppb*min and divided by the experiment duration of 60 minutes to get a value of ppb VOC over time. This was converted to mol C cm$^{-3}$ assuming each molecule had 8 carbons. Using these two numbers for dissolved and gas-phase organic carbon, we calculate that <0.001% of dissolved organic carbon goes to the gas-phase during the experiment.

L291: "Utilizing $\Delta$VOC over 60 minutes, a DOC measurement of 32 µM, and assuming 8 carbons based on the median carbon number of molecules in these experiments (Section 3.4), we calculate the fraction of organic carbon going to the gas-phase in the experiment to be <0.001%."

L355: "..based on standard deviation of O3 mixing ratios measured in 2019" -- is the data normally distributed? Is this an appropriate measure of your uncertainty?

Below is a histogram of the O$_3$ mixing ratios measured in 2019, showing the 1$\sigma$ and 2$\sigma$ standard deviations. The data is normally distributed and we think standard deviation of O$_3$ is an appropriate measure of uncertainty.

[Figure]

L356: why not include these additional uncertainties, or an estimate, to provide a more true measure of the uncertainty in your field VOC yield?

Remaining uncertainties are from the flux measurements due to sensor noise and sampling uncertainty due to the natural variability in turbulence. We conservatively estimate the combined flux uncertainties from these two factors to be 60% but note that these are hard to accurately quantify. Therefore, we do not propagate these numbers through the yields as we are less certain in this flux uncertainty than we are in the O$_3$ uncertainty. Addition of this uncertainty in the field yields would still result in the total field VOC yield less than the total lab VOC yield at the high

end. At the low end, the total field VOC yield would be even lower. The same potential causes for this discrepancy between field and lab VOC yields could still be applied.

L378: "The range in $\varphi_{VOC,field}$ reported below is based on standard deviation of $O_3$ mixing ratios measured in 2019. However, we note that the uncertainty in calibration factors (Sect. S1) and VOC fluxes would drive a much larger uncertainty than just $O_3$ variability. This additional uncertainty from VOC fluxes is estimated at 60% from the combination of sensor noise and sampling uncertainty."

L376-378: "any O3 source variability was not directly measured simultaneously during experiments, meaning that quicker O3 fluctuations than what occur in the ambient could have heightened laboratory yields." -- do the authors expect this is a significant source of higher VOC yield when delta-VOC mirrors delta-O3? If so, this should be discussed further.

This was included to think through all potential differences between the field and laboratory experiments. We do not think this was a significant source of higher VOC yield. The lamp used to generate $O_3$ should be stable on the time scale of the experiments and decreased intensities are <15% per 1000 hours of continuous operation. Additionally, the mass flow controllers regulating gas flow through the lamp have settling times of less than two seconds.

L380: "[I-] a factor of three lower" -- clarify if this is measured or inferred.

Thanks for pointing this out, we realized we did not state that [I⁻] was measured during the field study as well. The measured value was 42 ± 5.3 nM on average. We have updated the text:

L409: "We compare lab data to field data that has a factor of three lower [I⁻] (measured 42 ± 5.3 nM)"

L385: "caution our ability" -- should be "complicate our ability"?

We have taken your suggestion, the text now reads: " complicate our ability"

L405: (Figure 5 caption) Regression should be relationship, or similar? You are not quantitatively assessing the relationship between two variables (i.e., you are not applying a regression model)

Thanks, this is a good point. The text now reads: " Relationship between VOC yields"

L440: "a couple of hundred" -- avoid vague language, and give the range from the literature you cite and their uncertainties.

Thanks, there is spatiotemporal variability in these numbers which give them a range. We have removed a couple of hundred and replaced with <300 ppt. Updated text is here:

L473: "less than 300 ppt in coastal areas (Novak et al., 2022;  Shaw et al., 2010)"

Author note:

After submitting the preprint of this paper, we realized we had not commented that our observations of aldehydes from a coastal site were consistent with other coastal measurements influenced by macroalgae (Tokarek et al., 2019), not just specific to ozonolysis experiments. We have added a line in the paper:

L445: "The findings in this work are consistent with prior observations of $C_8$-$C_{10}$ aldehydes in coastal regions influenced by macroalgal species (Tokarek et al., 2019)."

References

Abbatt, J. P. D. and Ravishankara, A. R.: Opinion: Atmospheric multiphase chemistry – past, present, and future, Atmospheric Chemistry and Physics, 23, 9765–9785, https://doi.org/10.5194/acp-23-9765-2023, 2023.

Halsey, K. H. and Giovannoni, S. J.: Biological controls on marine volatile organic compound emissions: A balancing act at the sea-air interface, Earth-Science Reviews, 240, 104360, https://doi.org/10.1016/j.earscirev.2023.104360, 2023.

Hulswar, S., Simó, R., Galí, M., Bell, T. G., Lana, A., Inamdar, S., Halloran, P. R., Manville, G., and Mahajan, A. S.: Third revision of the global surface seawater dimethyl sulfide climatology (DMS-Rev3), Earth System Science Data, 14, 2963–2987, https://doi.org/10.5194/essd-14-2963-2022, 2022.

Lana, A., Bell, T. G., Simó, R., Vallina, S. M., Ballabrera-Poy, J., Kettle, A. J., Dachs, J., Bopp, L., Saltzman, E. S., Stefels, J., Johnson, J. E., and Liss, P. S.: An updated climatology of surface dimethlysulfide concentrations and emission fluxes in the global ocean, Global Biogeochemical Cycles, 25, https://doi.org/10.1029/2010GB003850, 2011.

Magi, L., Schweitzer, F., Pallares, C., Cherif, S., Mirabel, P., and George, C.: Investigation of the Uptake Rate of Ozone and Methyl Hydroperoxide by Water Surfaces, 101, 4943–4949, https://doi.org/10.1021/jp970646m, 1997.

Moreno, C. G., Gálvez, O., López-Arza Moreno, V., Espildora-García, E. M., and Baeza-Romero, M. T.: A revisit of the interaction of gaseous ozone with aqueous iodide. Estimating the contributions of the surface and bulk reactions, Phys. Chem. Chem. Phys., 20, 27571–27584, https://doi.org/10.1039/C8CP04394A, 2018.

Pagonis, D., Sekimoto, K., and de Gouw, J.: A Library of Proton-Transfer Reactions of H3O+ Ions Used for Trace Gas Detection, Journal of The American Society for Mass Spectrometry, 30, 1330–1335, https://doi.org/10.1007/s13361-019-02209-3, 2019.

Peacock, M., Freeman, C., Gauci, V., Lebron, I., and Evans, C. D.: Investigations of freezing and cold storage for the analysis of peatland dissolved organic carbon (DOC) and absorbance properties, Environ. Sci.: Processes Impacts, 17, 1290–1301, https://doi.org/10.1039/C5EM00126A, 2015.

Prophet, A. M., Polley, K., Van Berkel, G. J., Limmer, D. T., and Wilson, K. R.: Iodide oxidation by ozone at the surface of aqueous microdroplets, Chem. Sci., 15, 736–756, https://doi.org/10.1039/D3SC04254E, 2024.

dos Santos, D. J. V. A., Müller-Plathe, F., and Weiss, V. C.: Consistency of Ion Adsorption and Excess Surface Tension in Molecular Dynamics Simulations of Aqueous Salt Solutions, J. Phys. Chem. C, 112, 19431–19442, https://doi.org/10.1021/jp804811u, 2008.

Schneider-Zapp, K., Salter, M. E., Mann, P. J., and Upstill-Goddard, R. C.: Technical Note: Comparison of storage strategies of sea surface microlayer samples, Biogeosciences, 10, 4927–4936, https://doi.org/10.5194/bg-10-4927-2013, 2013.

Shaw, M. D. and Carpenter, L. J.: Modification of Ozone Deposition and $I_2$ Emissions at the Air–Aqueous Interface by Dissolved Organic Carbon of Marine Origin, Environ. Sci. Technol., 47, 10947–10954, https://doi.org/10.1021/es4011459, 2013.

Shi, H. and Adams, C.: Rapid IC–ICP/MS method for simultaneous analysis of iodoacetic acids, bromoacetic acids, bromate, and other related halogenated compounds in water, Talanta, 79, 523–527, https://doi.org/10.1016/j.talanta.2009.04.037, 2009.

Tokarek, T. W., Brownsey, D. K., Jordan, N., Garner, N. M., Ye, C. Z., and Osthoff, H. D.: Emissions of C9 – C16 hydrocarbons from kelp species on Vancouver Island: Alaria marginata (winged kelp) and Nereocystis luetkeana (bull kelp) as an atmospheric source of limonene, Atmospheric Environment: X, 2, https://doi.org/10.1016/j.aeaoa.2019.100007, 2019.